# Synthesis and Characterization of Room Temperature Vulcanized Silicone Rubber Using Methoxyl-Capped MQ Silicone Resin as Self-Reinforced Cross-Linker

**DOI:** 10.3390/polym11071142

**Published:** 2019-07-03

**Authors:** Jianye Ji, Xin Ge, Xiaoyan Pang, Ruoling Liu, Shuyi Wen, Jiaqi Sun, Weijie Liang, Jianfang Ge, Xunjun Chen

**Affiliations:** 1Guangdong Engineering Research Center of Silicone Electronic Fine Chemicals, College of Chemistry and Chemical Engineering, Zhongkai University of Agriculture and Engineering, Guangzhou 510225, China; 2School of Materials and Energy, Guangdong University of Technology, Guangzhou 510006, China; 3School of Materials Science and Engineering, Northwestern Polytechnical University, Xi’an 710072, China

**Keywords:** cross-linker, MQ silicone resin, hydrosilylation, silicone rubber, modified

## Abstract

Methoxyl-capped MQ silicone resin (MMQ) was first synthesized by the hydrosilylation of vinyl-containing MQ silicone resin and trimethoxysilane and then used in condensed room-temperature vulcanized (RTV) silicone rubber as a self-reinforced cross-linker. Results show that modified silicone rubber exhibits good light transmission. Compared with unmodified silicone rubber, the hardness, tensile strength and elongation of MMQ at the break are increased by 26.4 A, 2.68 MPa and 65.1%, respectively. In addition, the characteristic temperature of 10% mass loss is delayed from 353.5 °C to 477.1 °C, the temperature at maximum degradation rate is also delayed from 408.9 °C to 528.4 °C and the residual mass left at 800 °C is increased from 1.2% to 27.7%. These improved properties are assigned to the synergistic effect of the rigid structure of MMQ, the formation of a dense cross-linking structure in polymers and the uniform distribution of MMQ cross-linking agent in RTV silicone rubber.

## 1. Introduction

As an industrial product which possesses the characteristics of inorganic/organic material components, silicone rubber has been widely applied in the aerospace, cable, automotive, medical appliances and electronics industries fields, among others, due to its peculiar physical and chemical performance, such as its high temperature resistance [1,2], weather resistance [3], excellent electrical insulation and good chemical stability [4]. Although silicone rubber offers many advantages not possessed by other organic rubbers, the pure-curing silicone rubber is not able to meet the requirements of practical applications, because the weak interaction and high flexibility of the polydimethylsiloxane main chains results in poor mechanical properties of the vulcanized silicone rubber [5]. Therefore, the mechanical properties of silicone rubber should be improved to increase its practical application value and further broaden its scope of application.

In general, the mechanical properties of silicone rubber can be improved through the modification of the polydimethylsiloxane (PDMS) matrix; for instance, the introduction of phenyl, phenylene, vinyl groups or other organic groups to the PDMS main chain [6]. In addition, the incorporation of fillers is another effective approach to enhancing the properties of silicone rubber, such as by using montmorillonite, modified montmorillonite, carbon nanotubes, pyrogenic silicas, nano alumina and graphene as fillers to enhance the mechanical performance of silicone rubber [7,8]. However, it is noteworthy that despite the fact that the adoption of these fillers possibly improves the mechanical properties of silicone rubber, these fillers are difficult to disperse uniformly in the PDMS matrix, which results in increased system viscosity. Moreover, the silicone rubber enhanced by these fillers usually possesses a low light transmittance due to the poor system compatibility of fillers and the PDMS matrix, which will severely limit its application in areas with high optical requirements [9], such as in the fields of light emitting diode (LED) encapsulation and optical bonding. Besides this, using a novel cross-linker to reinforce the properties of silicone rubber is also effective; for instance, using α,ω-bis(trimethox-ysilane)-polydimethylsiloxane as a cross-linker improves the mechanical properties of silicone rubbers by increasing its cross-linking density [10]. It has also been reported that polymethyl-methoxysiloxane (PMOS) as a cross-linker can improve the thermal and mechanical properties of the siloxane elastomer due to its in-situ microscale improvement effect. It has been found that the tensile stress and elastic module of the siloxane elastomer were increased two and three times, respectively, when the PMOS cross-linking agent increased from 15.1 wt % to 41.6 wt % [11]. However, it is anticipated that these linear polymethylsiloxy-type cross-linking agents will have excellent compatibility with the PDMS matrix, but the improvement of the mechanical properties of silicone rubber is still limited due to the weak interaction in the cross-linking agent molecular chain.

Recently, Li and his co-workers prepared rosin-modified aminopropyltriethoxysilane as a cross-linking agent applied to condensed silicone rubber, showing that the thermal stability and mechanical properties of silicone rubber were improved to some extent [12]. However, it cannot be ignored that the introduction of a macromolecular phenanthrene ring structure will reduce the compatibility of the system and result in the phase separation of the polymer system, as well as strongly reducing the transparency of the silicone rubber. Simultaneously, the mechanical properties of silicone rubber are also enhanced finitely. The above-mentioned factors show that modified silicone rubber is still a long way from practical application, especially in areas with high optical requirements. Some research shows that using polyhedral oligomeric silsesquioxanes (POSS) and MQ silicone resin-type organic–inorganic hybrid materials as cross-linking agents can remarkably improve the thermal and mechanical properties of silicone rubbers due to their strongly rigid and three-dimensional spatial structures and good compatibility with the PDMS polymer [13,14,15]. Moreover, the light transmittance of silicone rubber is also at a high level, which greatly enlarges its application areas.

MQ silicone resins are polymers consisting of mono-terminated siloxane (R_3_SiO_1/2_, the M units) and tetra-terminated siloxane (SiO_4/2_, the Q units), wherein R is a functional or nonfunctional organic group, such as hydrogen, vinyl, methyl, aryl and some related organic or inorganic groups [16]. Its unique organic/inorganic structure makes it have good compatibility with the PDMS matrix, and it is very suitable for preparing high-performance and good light-transmittance silicone rubber.

In order to achieve a better reinforcement effect, it is necessary to maintain the chemical bonding point between the resin and the silicone rubber while the MQ silicone resins are used to reinforce condensed room-temperature vulcanized (RTV) silicone rubber [17]. The conventional method is to retain a portion of the silicon hydroxy group without being capped during the synthesisis of the MQ silicone resin and then prepare the RTV silicone rubber by the dehydration condensation of the residual silanols and hydroxy-terminated polydimethylsiloxane (HPDM S). However, the silanol-containing MQ silicone resin may be subject to further polymerization, which may affect the storage stability of the resin. Meanwhile, the hydroxyl content is difficult to control in the process of synthesizing MQ silicone resin, meaning that the reinforcement of silicone rubber cannot obtains stable performance. For these reasons, the reliability of the preparation and application of the silicone rubbers is greatly limited.

An effective way to improve the storage stability and application reliability is by using alkoxy-modified MQ silicone resin [18]. When the modified MQ silicone resin is applied to reinforce condensation RTV silicone rubber, the Si-OCH_3_ groups provide a cross-linking site for HPDMS polymer, and the Si-O backbone chains acts as a reinforcing domain. The modified silicone rubber will possess excellent mechanical properties and good light-transmitting properties, which give it a higher value and wider range of application.

At present, few studies on the application of methoxyl-capped MQ silicone resin (MMQ) as a self-reinforced cross-linking agent for condensed RTV silicone rubber have been published. Regarding this question, an approach to prepare MMQ and explore its effect on condensed RTV silicone rubber is reported. In this work, MMQ was prepared by the hydrosilylation of vinyl–MQ silicone resin and trimethoxysilane (TMOS). The condensed RTV silicone rubber was prepared through hydrolysis under moisture and condensation between the MMQ cross-linker and HPDMS at room temperature in the presence of an organotin catalyst. The effect of MMQ on the cross-linking density, morphology, hydrophobic property, transparency, mechanical and thermal stability properties of the RTV silicone rubber were all explored. It is expected to obtain an RTV silicone rubber material with strong hydrophobicity, good light transmittance, excellent mechanical and thermal stability properties, and to further extend its application range.

## 2. Materials and Methods

### 2.1. Materials

Vinyl–MQ silicone resin (VMQ, M/Q = 0.8, vinyl content 4.0 wt %) was purchased from Jinan Xinshuo Chemical industry Co., Ltd. (Jinan, China); trimeth-oxysilane (TMOS, AR) and tetraethyl orthosilicate (TEOS, AR) were purchased from Shanghai Macklin Biochemical Co., Ltd. (Shanghai, China); chloroplatinic acid (H_2_PtCl_6_, 5000 ppm, AR) was provided by Tianjin Maisike Chemical industry Co., Ltd. (Tianjin, China); dibutyltin dilaurate (DBTDL, 98 wt %, AR) was obtained from Jilin Huaxin Chemical Co., Ltd. (Jilin, China); hydroxy-terminated polydimethylsiloxane (HPDMS, 20,000 cst) was purchased from Hubei Xinsihai Chemical Co., Ltd. (Zaoyang, China); ethyl acetate, perchloric acid, potassium hydroxide, absolute ethanol, cresol red and thymol blue were all obtained from Shanghai Aladdin Biochemical Technology Co., Ltd. (Shanghai, China).

### 2.2. Synthesis of the MMQ

MMQ was prepared by the hydrosilylation reaction of VMQ and TMOS. Toluene (9.0 mL), VMQ (50.0 g) and H_2_PtCl_6_ (100 uL) were added into a 250 mL three-necked flask. Then, TMOS (10.9 g) was added into this flask drop by drop with continued stirring at 70 °C for 5 h under a nitrogen atmosphere. Finally, the polymer–toluene solution was distillated at 100 °C for 2 h to obtain the polymer.

### 2.3. Synthesis of the Condensed RTV Silicone Rubber

In general, condensed RTV silicone rubbers are composed of HPDMS, a cross-linker, curing catalyst, enhancing filler, colorant, etc, and each component will affect the comprehensive performance of the RTV silicone rubber. In this study, the concentration of curing catalysis, the type of HPDMS, the total reactive group content of cross-linker in the polymer system and the reaction condition were kept constant. The RTV silicone rubber with different amounts of loading of MMQ cross-linking agent were prepared via hydrolysis and condensation under moisture between the cross-linking agent and HPDMS in the existence of a DBTDL catalyst. The amounts of RTV silicone rubber raw materials are listed in Table 1.

Taking a preparation of MMQ-1 as a sample, MMQ (5.87 g), HPDMS (50.0 g) and the curing catalyst were added into a 250 mL three-necked flask, and then vigorously stirred for 30 min under a nitrogen atmosphere to mix homogeneously. Then, the volatile compounds were removed under vacuum around 20 min. Finally, the mixture was quickly poured into a polytetrafluoroethylene mold. RTV silicone rubber with a smooth surface was obtained after curing for about 72 h at room temperature. All the RTV silicone rubber samples were also prepared under the same conditions.

### 2.4. Characterizations and Measurements

The structure of VMQ and MMQ specimens were characterized by VERTEX70 Fourier transform infrared (FT-IR) spectra (Shimadzu Corporation, Kyoto, Japan) combined with the HF-7 detachable liquid pool (Bruker Corporation, Karlsruhe, Germany) with a capacity of 1 mL. The IR spectra of silicone rubber samples were obtained with a Pike Miracle spectrometer by ATR-IR (attenuated total reflectance infrared spectroscopy).

1H NMR spectra for VMQ and MMQ were recorded at 25 °C on an Av 500 Hydrogen Nuclear Magnetic Resonance Spectrometer (Bruker Corporation, Karlsruhe, Germany) at frequencies of 500 MHz using deuterated chloroform (CDCl_3_) as the solvent and tetramethylsilane (TMS) as an internal standard.

The surface morphologies of the RTV silicone rubber samples were studied using an EVO 18 (Carl Zeiss, Jena, Germany) electron microscope at a voltage of 15 kV.

The contact angles of the samples were measured by the JC 2000A automatic contact angle meter (Shanghai Zhongzheng Chemical Technology Co., Ltd., Shanghai, China).

Transmittance spectra were received with Shimadzu UV-1800 (Shimadzu Corporation, Kyoto, Japan), in a range from 300 nm to 800 nm, and air was used as the reference.

The mechanical properties were determined with an AGS-J universal testing machine (Shimadzu Corporation, Kyoto, Japan). A cross-head speed of 50 mm·min^−1^ and a load of 100 N were used during the testing, and the average of five-fold measurements was calculated.

The hardness of the samples was measured using the LX-A durometer (Ji Tai Keyi Technology Co., Ltd., Shenzhen, China) at a temperature of 25 °C and relative humidity (RH) of 50%.

The dynamic mechanical properties of the silicone rubbers were carried out with a DMA-1 (Mettler Toledo Instrument Co., Ltd., Shanghai, China), and a frequency of 1 Hz and heating rate of 3 °C·min^−1^ were used, with scanning from −135 °C to −60 °C per sample.

Thermogravimetric analysis (TGA) measurement was carried out on Mettler Toledo TG thermal analyzer (Mettler-Toledo AG Corporation, Columbus, OH, USA) at a flow rate of 20 mL/min in nitrogen; the samples were heated from 40 °C to 800 °C at a rate of 10 °C /min.

### 2.5. The Conversion of Vinyl in VMQ

The conversion of vinyl in VMQ was measured by iodometry. Firstly, the MMQ sample was dissolved into a 250 mL Erlenmeyer flask containing 20.0 mL of the CCl_4_ and shaken evenly. Then, 10.0 mL of IBr solution was added, shaken evenly and placed in a shadowy place for 1 h. After that, 5.0 mL of KI solution was introduced to the flask and shaken for about 3 min. Then, 40.0 mL deionized water was added. Finally, using 0.1 mol/L of Na_2_S_2_O_3_ to titrate the solution until it turned pale yellow, 2.0 mL of 1 wt % soluble starch solution was added, and the solution was continuously titrated until the blue disappeared. The conversion of vinyl in VMQ is determined by the following equation:(1)$$\Phi =\frac{27\times (V_{0}-V_{1})\times 0.001\times 0.5\times C_{{\mathrm{Na}}_{2}S_{2}O_{3}}}{M}\times 100\%$$
(2)$$\Psi =1-\frac{\Phi }{4.0\%}\times 100\%$$
where 27 (g/mol) is the molar mass of vinyl; 0.001 is the unit conversion constant between mL with L; 0.5 is the coefficient of mole balance; Φ is the mass fraction of vinyl in MMQ (%); V_0_ is the volume of Na_2_S_2_O_3_ consumed by the blank experiment; and V_1_ is the volume of Na_2_S_2_O_3_ consumed by the experimental group. C_Na2S2O3_ is the concentration of Na_2_S_2_O_3_ (mol/L); M is the mass of MMQ samples; Ψ is the conversion of vinyl in VMQ; and 4.0% is the mass fraction of vinyl.

### 2.6. Mass Fraction of Methoxy Groups in MMQ

The mass fraction of methoxy groups in MMQ was measured by a method according to a procedure in the literature with a small amendment [19], and the perchloric acid acetylation reagent, sodium hydroxide and indicator were prepared according to the literature [6]. Firstly, the MMQ sample was dissolved into a 250 mL Erlenmeyer containing 5 mL of the perchloric acid acetylation reagent and with a standing time of the solution of 7 min. Secondly, 1.8 mL pure water was added into the Erlenmeyer flask and shaken evenly; then, 10 mL mixture (the volume ratio of pyridine to pure water is 3/1) was added, shaken evenly and hydrolyzed at room temperature for 30 min. Then, 50 mL ethanol was introduced to dilute the hydrolysate. Thirdly, 5 drops of indicator were added into the Erlenmeyer and shaken for about 1 min. Finally, the solution was continuously titrated with potassium hydroxide ethanol solution until a red coloration appeared. The mass fraction of the methoxyl groups in MMQ was determined by the following equation:(3)$$\omega =\frac{(V_{2}-V_{3})\times 0.001\times C_{\mathrm{KOH}}\times 31.03}{m}\times 100\%$$
where ω is the mass fraction of methoxy groups in MMQ (%); C_KOH_ is the concentration of KOH (mol/L); m is the mass of MMQ; V_3_ is the volume of KOH consumed by experimental group; and V_2_ is the volume of KOH consumed by the blank experiment. The constant 31.03 is the molar mass of the methoxy group.

### 2.7. Swelling Experiment

The density of RTV silicone rubbers can be estimated by extraction and swelling experiments. The experiment proceeded as follows: about 1 cm × 1 cm of the sample was cut from a 3 mm thick sheet and weighed as W1. After that, the samples were immersed in a bottle containing 10.0 mL toluene with a piston for 168 h, and the toluene was renewed per day. Then, the sample was taken out of the bottle, and the weight (W_sw_) of the swollen samples were determined after the solvent on the surface of the sample was blotted. Finally, the swollen samples were placed in a vacuum oven to a constant weight (W_0_). The soluble fraction of the samples was determined by the following equation:(4)$$W=\frac{w_{1}-w_{0}}{w_{1}}\times 100\%$$
where W is the soluble fraction (%). The degree of swelling (η_sw_, %) of the samples was also determined by the following equation:(5)$$\eta_{\mathrm{sw}}=\frac{w_{\mathrm{sw}}-w_{0}}{w_{0}}\times 100\%$$

## 3. Results and Discussion

### 3.1. Structural Analysis

The MMQ was synthesized first by the hydrosilylation of VMQ and TMOS using H_2_PtCl_6_ as a reaction catalyst. The anti-Martensii addition reaction was the chief method used in this hydrosilylation [20], which was also confirmed by 1H-NMR spectroscopy analysis. Theoretically, this reaction of Si–CH=CH_2_ and Si–H is in the molar ratio of 1/1, but to overcome the side effect of the lively Si–H, and the steric hindrance of VMQ which wasmainly caused by vinyl–MQ cage and reacted vinyl groups during further addition reaction, the excess molar ratio of TMOS to VMQ was used to promote the reaction more completely, and the choice of the molar ratio of Si–H/Si–CH=CH_2_ was 1.05/1 in this experiment. Finally, by chemical titration, the conversion of vinyl in VMQ and the content of methoxy groups in MMQ were measured and calculated, respectively, and the result showed that the conversion of vinyl in VMQ was 75.7%, and the mass fraction of methoxy groups in MMQ was 3.48%, which amount to 1.12 millimoles of methoxy group per gram of MMQ.

Then, the condensed RTV silicone rubbers were prepared via the hydrolytic/condensation reaction of the cross-linker and HPDMS polymer in the existence of DBTDL catalysis under moist conditions at room temperature. Due to this curing reaction being sensitive to the temperature and the relative humidity of the ambient environment, the curing mold was placed on the water bath, the temperature of the water bath was controlled at 25 °C and the constant relative humidity of the environment was maintained. In addition, a small amount of methanol was produced during the curing process, meaning that the reaction should be carried out in a ventilated environment.

To explore the effects of different amounts of MMQ on the properties of the condensed RTV silicone rubbers, a series of RTV silicone rubber samples were prepared by the introduction of the same methoxyl content of MMQ and the same total amounts of reactive groups of the cross-linking agent in the system were controlled. The prepared route of RTV silicone rubber is shown in Figure 1.

The FT-IR spectra of VMQ (a), MMQ (b), HPDMS (c) and the cured PDMS (d) are shown in Figure 2. As can be seen, the infrared spectra of MMQ was analogous to that of VMQ, which indicates that MMQ and VMQ possess similar structures. In the curve of VMQ, the sharp band at 3050 cm^−1^ is the stretching vibration absorption of C–H in Si–CH=CH_2_. The peaks at 2960 cm^−1^ and 2910 cm^−1^ are assigned to the stretching vibration absorption of C–H in –CH_3_. The peaks at 1600 cm^−1^, 1410 cm^−1^ and 960 cm^−1^ corresponded to the characteristic peak of the vinyl group [21]. The peaks at 1260 cm^−1^ and 852 cm^−1^ are ascribed to the scissoring vibration absorption of Si–CH_3_ [22]. The broad absorption peak at 1100–1080 cm^−1^ is ascribed to the stretching vibration absorption of Si–O–Si [23]. Compared with the curve of a, the symmetric stretching vibrations of C–H on Si–CH_2_–CH_2_–Si formed during the hydrosilylation reaction appear at 2850 cm^−1^ [24], and the characteristic absorption peak of vinyl at 1600 cm^−1^ is significantly weakened [25]. Moreover, although the absorption peak of Si–OCH_3_ could not be observed at 1000–1100 cm^−1^ due to the shielding effect of the Si–O–Si absorption peak, the absorption peak of Si–O–Si became stronger than that of VMQ under the synergistic effect of Si–OCH_3_. These characteristics indicated that MMQ was synthesized successfully. In addition, in the curve of c, a relatively broad absorption peak at 3440 cm^−1^ and a small sharp peak at 1660 cm^−1^ are ascribed to the stretching and deformation vibration of H-bonded silanol (Si–OH) group in HPDMS, respectively [26]. Compared with the infrared spectra of HPDMS (curve c), it can be found that the characteristic absorption peak of Si–OH at 3440 cm^−1^ and 1660 cm^−1^ disappeared completely in the curve of d; besides this, an absorption peak at 2850 cm^−1^ attributed to C–H on Si–CH_2_–CH_2_–Si and a characteristic peak of Si–CH=CH_2_ at 1600 cm^−1^ introduced by MMQ appeared. Meanwhile, the asymmetric stretching vibration band of Si–O–Si around 1010 cm^−1^ is slightly wider than that of HPDMS, with the peak maybe resulting from the random structure of the absorption peak of Si–O–Si becoming slightly stronger than that of HPDMS. These characteristics indicated that a new network of Si–O–Si was formed in the PDMS matrix, and that the condensed RTV silicone rubber was prepared successfully.

The structures of VMQ and MMQ were also confirmed by 1HNMR spectra, as shown in Figure 3a,b. The characteristic peaks at 0–0.17 ppm and 5.56–6.12 ppm corresponded to the protons on –CH3 and –CH=CH_2_ groups, respectively, for M units in VMQ. Compared with VMQ, except for the chemical shifts of proton peaks of –CH_3_ and –CH=CH_2_ groups, the characteristic signal located at about 1.0–1.3 ppm was attributed to the proton peaks of C–CH_3_. The characteristic peak at 1.68–1.84 ppm corresponded to the proton of tertiary carbon (C–H). The strong signal was located at 0.66–0.79 ppm for the protons of the Si–CH_2_CH_2_–Si group, which indicates that the hydrosilylation of VMQ and TMOS was carried out as a result of α-addition products and β-addition products and dominated primarily by β-addition [27]. In addition, the characteristic signal located at 3.6–3.8 ppm was assigned to the proton of Si–OCH_3_, and the chemical shift of Si–H occurring at 4–5 ppm was not found. These analyses indicate that MMQ was synthesized successfully.

### 3.2. Morphology of the Silicone Rubber

It can be observed in Figure 4a that the sample using TEOS as an individual cross-linker is completely transparent. The samples still remain transparent (MMQ-1~MMQ-4) or translucent (MMQ-5) after introducing MMQ, which indicated that the MMQ had good compatibility with the PDMS matrix, and the microphase separation of the polymers was not obvious. In addition, scanning electron microscopy was used in observing the morphology of RTV silicone samples. The result indicated that with the increased amount of loading of MMQ, the micron-sized spherical domains (from Figure 4b–g) in polymers also increased gradually. Some clear spherical accumulation domains can be observed in the surface of silicone rubber when the percentage of the MMQ reaches 50%. These aggregation domains could be the rich clusters of MMQ, which are due to the aggregation of self-cross-linked MMQ cross-linker [28]. Therefore, it is necessary to disperse the MMQ well in the polymer matrix before the curing catalyst is added to avoid the formation of these aggregations. Meanwhile, maintaining the proper amount of MMQ addition is another consideration.

### 3.3. Hydrophobicity of the RTV Silicone Rubber

The hydrophobicity of the RTV silicone rubber can be evaluated by the water contact angle (θ) on the sample surface; when the numeric value of θ is less than 90, the surface can be considered hydrophilic, and when the numeric value of θ is greater than 90, the surface can be considered hydrophobic [29]. The effect of different amounts of MMQ cross-linking agent on hydrophobicity of the silicone rubbers was investigated. As presented in Figure 5, compared with the contact angle of sample MMQ-0 (55.37°), the water contact angle of all the samples using MMQ as cross-linker was increased. This may be attributed to the formation of a denser three-dimensional network structure and the decrease of the sample surface energy, because, with the increased amounts of MMQ, the chain entanglements and intermolecular cohesion of polymer also increased, which restricted the motion of molecular chain and reduced the distance of molecular network [30]. Moreover, the PDMS molecular chain migrated to the molecular interior after hydrolysis condensation, and the hydrocarbon groups migrated to the molecular surface, which reduced the surface energy of RTV silicone rubber and improved the hydrophobicity of the sample. For these reasons, the contact angle of MMQ-5 should be less than that of MMQ-4 due to the non-uniform distribution caused by the self-cross-linking and agglomeration of MMQ in the polymer matrix. However, the results indicated that the average value of the contact angle of MMQ-5 was up to 120.12° and was obviously higher than the value of 101.55° for MMQ-4. This is likely due to the excessive addition of MMQ resulting in an uneven dispersion in the matrix, thus causing a rougher surface of the sample. When in contact with the water droplets, a phenomenon similar to the “lotus effect” is formed, making the hydrophobicity even higher [31].

### 3.4. Cross-Linking Density of the RTV Rubber

A process of swelling will occur when a high amount of polymer is in contact with a related solvent. The polymer cannot be dissolved directly but swells by absorbing the solvent due to the cross-linking network structure [32]. A higher cross-linking density of the polymer results in greater hindrance of infiltration and diffusion of the solvent molecule, as well as lower polymer solubility. Therefore, the cross-linking density of silicone rubber can be estimated by measuring their dissolving ratio and swelling. The dissolving ratio and swelling results of RTV silicone rubbers were evaluated as illustrated in Figure 6e,f. It can be found that the soluble fraction and the degree of swelling of samples declined obviously after using MMQ as cross-linking agent. This is attributed to the incorporation of MMQ causing the formation of a denser three-dimensional network structure that effectively limited the movement of the flexible chain in polymers and reduced the molecular distance of the polymers, which hindered the infiltration of the solvent molecule [33]. With the increased addition of MMQ cross-linking agent, the soluble fraction and the degree of swelling of samples decreased continually. However, when the MMQ cross-linker content reached 50 wt %, the soluble fraction of the silicone rubber increased. This may be due to the self-cross-linking and aggregation from an overdose of MMQ cross-linker, which results in heterogeneous MMQ-rich domains [34]. Simultaneously, the self-cross-linking and aggregation of MMQ cross-linker also led to the decrease of active sites, which caused a decrease of the cross-linking density, and then induced increases of the soluble fraction of the silicone rubber. When the MMQ cross-linker content was 40 wt % (MMQ-4), the silicone rubber remained at a relatively low soluble fraction and degree of swelling. This indicated that silicone rubber possesses a uniform and compact cross-linking structure; compared with the sample that only used TEOS as cross-linking agent, the soluble fraction and the degree of swelling of the sample were decreased by 0.79 wt % and 52 wt %, respectively.

### 3.5. Mechanical Properties of RTV Silicone Rubber

Mechanical properties are the most important factors which directly influence the application of RTV silicone rubber. In order to investigate the effect of the MMQ cross-linking agent contents on the mechanical properties of RTV silicone rubber, the tensile properties, shore A hardness and dynamic mechanical property of RTV silicone rubbers were estimated.

The mechanical properties of the RTV silicone rubbers—for instance, the stress–strain curves, tensile strength, elongation at break, modulus and shore A hardness—are illustrated in Figure 6b–d,g, respectively. The tests for the tensile properties and shore A hardness of MMQ-0 and MMQ-4 distinctly show that RTV silicone rubber using MMQ as a cross-linker has improved mechanical properties. Specifically, the tensile strength of the RTV silicone increased from 0.43 MPa to 3.11 MPa and the elongation at break from 134.0% to 199.1% upon moving from MMQ-1 to MMQ-4, which are increases by 2.68 MPa and 65.1%, respectively. In addition, the elastic modulus increased from 0.32 MPa to 1.15 MPa on moving from MMQ-0 to MMQ-4. Moreover, the Shore A hardness is increased by 26.4 A, compared with sample MMQ-0. These increased mechanical properties of the RTV silicone rubber may be attributed to the introduction of MMQ which increases the content of effective rigid elements and facilitates dense entanglements of molecular chains in RTV silicone rubber [35]. Due to all the samples having an equal total amount of methoxyl grounds, it could be suggested that the mechanical property improvements of the RTV silicone rubbers are due to the synergistic effect of the increased hard phase density and cross-linking density working together in the polymer. However, the result also indicated that the mechanical performances of these samples decreased while the loading amount of MMQ reached 50% (MMQ-5). This may be ascribed to the agglomeration and self-cross-linking from an excess of MMQ cross-linker, which results in homogeneous MMQ-rich domains in the RTV silicone rubber [20,36], as demonstrated by SEM in Figure 4g. Simultaneously, the agglomeration and self-cross-linking of MMQ cross-linker also led to the decreases of active sites, which caused the reduction of cross-linking density. This shows that the reinforcement of the mechanical properties of RTV silicone rubber is mainly due to the synergistic effect of the rigid structure of the MMQ cross-linking agent, the formation of dense cross-linking density in polymers and the uniform distribution of the MMQ cross-linking agent.

### 3.6. Dynamic Mechanical Properties Analysis

The effect of the MMQ cross-linker contents on the mechanical properties of the RTV silicone rubbers were also investigated by dynamic mechanical analysis (DMA). The curves of the storage modulus (E’) and tanδ of the samples are shown in Figure 7a,b. We know that the cross-linking density and the chemical structure of RTV silicone rubber have important influences on its modulus. It can be found that the storage modulus of RTV silicone rubber increases from 4599 MPa (MMQ-0) to 8679 MPa (MMQ-4) with the increased content of MMQ cross-linker. This observation can be attributed to the increment of cross-linking density and the rigid molecular chain resulting in a considerable restriction of the migration of polysiloxane segments [37]. However, with ab excessive content of MMQ addition (MMQ-5), inhomogeneous MMQ-rich domains in the polymer were formed due to agglomeration and self-cross-linking from an excess of MMQ cross-linker during the curing process (supported by SEM in Figure 4g). Otherwise, these agglomerations and the self-cross-linking of MMQ cross-linker also resulted in a relatively low cross-linking density of the RTV silicone rubber due to the loss of some active sites. Therefore, the E′ value of modified silicone rubber was decreased somewhat because of the inhomogeneous MMQ-rich domains and the decrease of the cross-linking density [38].

The loss factor (tanδ) is quite sensitive to the structural transformation of the materials. The tanδ of the RTV silicone rubber as a function of temperature is shown in Figure 7b, and the glass transition temperature (*T*g) of the sample corresponds to the tan δ peak temperature. It is found that the *T*g values of all the RTV silicone rubber samples varied from −112.8 °C to −107.8 °C, and the modified samples using MMQ as cross-linking agent had a higher *T*g compared with the unmodified silicone rubber sample. This indicates that the formation of the rigid three-dimensional spatial structure can effectively increase the cross-linking density of the polymers and lower the mobility of polysiloxane backbone chains, resulting in the enhancement of the *T*g.

### 3.7. The Transmittance of RTV Silicone Rubber

The ultraviolet transmittance of the polymers is influenced by their structures and contained functional groups. If the polymer contains chromophoric groups, such as double bonds and triple bonds, absorption will occur when it comes in contact with light corresponding to the characteristic frequency, which would result in a decrease in light energy and transmittance [39]. Light transmittance is an important performance of silicone rubber that determines whether it can be used in the fields of LED packaging, optical component bonding or other fields with high optical requirements. Therefore, the effects of different loading amounts of MMQ on the light transmittance of RTV silicone rubbers were investigated. Figure 8a shows the transmittance spectra of RTV silicone rubber in the wavelength range from 300 nm to 800 nm, and Figure 8b shows the spectrogram in the wavelength range from 350 nm to 450 nm. It can be seen that the values of the ultraviolet transmittance of the sample without using MMQ as a cross-linker are the highest (93.9% for MMQ-0 at wavelength of 400 nm) among all those samples using MMQ as cross-linking agent (92.4% for MQMQ-1, 90.5% for MMQ-2, 87.7% for MMQ-3, 86.2% for MMQ-4 and 83.4% for MMQ-5) at a wavelength of 400 nm, which showed that the transparency of silicone rubbers using MMQ as a cross-linker decreased with an increasing loading amount of MMQ. The decreasing transparency of these samples was likely ascribed to three reasons; on the one hand, the RTV silicone rubbers using MMQ as a cross-linking agent possess a denser molecular structure and larger molecule size, which hindered the transmission of light [37]. On the other hand, the agglomeration and self-cross-linking of the MMQ cross-linker caused heterogeneity and more roughness of the polymer matrix, which results in the scattering of light due to the interface effect [40]. Then, the existence of unsaturated double bonds of MMQ cross-linker were introduced into the silicone rubber with MMQ. Thirdly, some unsaturated double bonds still exist in the MMQ cross-linker, which were introduced into the silicone rubber with MMQ. Due to the intrinsic frequency of these double bonds matching the frequency of ultraviolet light, the radiation energy of corresponding frequency light was absorbed when it came into contact with ultraviolet light; thus, the ultraviolet transmittance of the materials was reduced.

### 3.8. Thermal Stability of RTV Silicone Rubber

In order to investigate the effects of the MMQ cross-linker on the thermal stability of the RTV silicone rubber samples, their thermal degradation behavior was characterized by TGA in a nitrogen atmosphere. The thermogravimetric analyses (TGA) and differential thermal gravity (DTG) curves of the polymer are graphed in Figure 9a,b. It was found that the characteristic temperature of 10% mass loss was delayed from 353.5 °C (MMQ-0) to 477.1 °C (MMQ-4). However, it began to decrease when the content of MMQ exceeded 40 wt %, but the temperature of the 10% weight loss of all the samples was still higher than that of MMQ-0 due to the introduction of MMQ with a rigid macromolecule structure. The stable rigid macromolecule structure of MMQ increases the chain entanglements of the polymer, which results in a restriction of the molecular motion of the polysiloxane chain [20,36]. Theis may lead to the slowing down of the rearrangement of polysiloxane backbone chains and the formation of cyclic oligomers, thereby decreasing the degradation rate of the silicone rubber. However, the excess loading amount of MMQ (MMQ-5) causes the agglomeration and self-cross-linking of MMQ cross-linker (supported by SEM in Figure 4g), which causes the insufficient number of active sites to completely react with Si–OH groups of polysiloxane, resulting in the increase of free flexible chains, accelerating the degradation rate.

The temperature at the maximum degradation rate of the RTV silicone rubber was also delayed from 408.9 °C (MMQ-0) to 426.3 °C (MMQ-1), and with an increase in the amount of loading of MMQ cross-linking agent, which exhibited a similar trend of 10% weight loss with the increase of MMQ loading. The MMQ-4 sample possessed the highest value of mass loss temperature (528.4 °C for MMQ-4), and this value declined slightly when the amount loading of MMQ was excessive (515.3 °C for MMQ-5), as depicted in Table 2. This may be due to the weak intermolecular force, and the trace amount of silanol groups in the MMQ-0 sample that accelerated the decomposition of PDMS matrix, resulting in it degrading rapidly at 408.9 °C [12,41]. The introduction of MMQ cross-linker can severely restrict the migration of the polysiloxane molecular chains and prevents the rearrangement of polysiloxane backbone chains, hence restricting the polymer being degraded into cyclic oligomers. Thus, the greatest rate of the weight loss temperature of the sample increased to 528.4 °C. By the same token, the excess loading amount of MMQ (MMQ-5) is not conducive to further increasing the temperature at the maximum degradative rate of the silicone rubber samples due to the formation of an MMQ agglomeration that causes the increases of free flexible chains in the silicone rubber matrix, accelerating the degradation of the polymer. However, it can be found that the temperature of the maximum degradation rate of the MMQ-5 sample only dropped by 10.1 °C as compared with that of MMQ-4. This may be attributed to the residual vinyl of MMQ improving the thermal stability of the RTV silicone rubbers [20]. These vinyls are likely to be cross-linked into dense phases via the appearance of radical reactions in the high-temperature degration domain, which can also restrict the mobility of HPDMS chains and hinder the further formation of cyclic oligomers, reducing the degradation rate of the sample in some extent. Thus, although the stability of the sample decreases significantly due to the agglomeration of MMQ cross-linking agents, the temperature at the maximum degradation rate of the MMQ-5 sample is only slightly below that of MMQ-4 due to the formation of a higher content of the dense phase [30]. In addition, the sample cured without MMQ cross-linker degraded into cyclic oligomers with only 1.2% residual mass left at 800 °C. However, all the samples cured with MMQ cross-linking agent possessed more residual yield than that of the MMQ-0, at 15.9% for MMQ-1, 19.1% for MMQ-2, 21.9% for MMQ-3, 27.7% for MMQ-4 and 23.1% for MMQ-5.

## 4. Conclusions

Methoxy-capped MQ silicone resin was prepared successfully, first via the hydrosilylation of vinyl-containing MQ silicone resin and trimethoxysilane, which was used as a cross-linker applied to reinforce the PDMS matrix. The obtained MMQ was then hydrolyzed and condensed with HPDMS in the presence of a curing catalyst under moisture at room temperature to prepare the condensed RTV silicone rubber. The hydrophobicity, transmittance, thermal and mechanical properties of the RTV silicone rubber were investigated. The results indicated that the transmittance of the modified silicone rubber remained at a high level (86.2% for MMQ-4 at a wavelength of 400 nm), amd the hydrophobicity, thermal gradation and mechanical properties of RTV silicone rubber were remarkably improved by using MMQ as a cross-linker. The contact angle increased from 55.73° to 101.55° upon moving from MMQ-1 to MMQ-4 compared with the unmodified sample, MMQ-0. The hardness, tensile strength and the elongation at the break of MMQ-4 were increased by 26.4 A, 2.68 MPa and 65.1%, respectively. In addition, the characteristic temperatures of 10% mass loss were delayed from 353.5 °C (MMQ-0) to 477.1 °C (MMQ-4), the temperature at maximum degradation rate is also delayed from 408.9 °C (MMQ-0) to 528.4 °C (MMQ-4) and the residual mass left at 800 °C is increased from 1.2% to 27.7%. This improved performance is ascribed to the synergistic effect of the rigid three-dimensional spatial structure of MMQ, the formation of a dense cross-linking structure in polymers and the uniform distribution of the MMQ cross-linking agent.

## Figures and Tables

**Figure 1 polymers-11-01142-f001:**
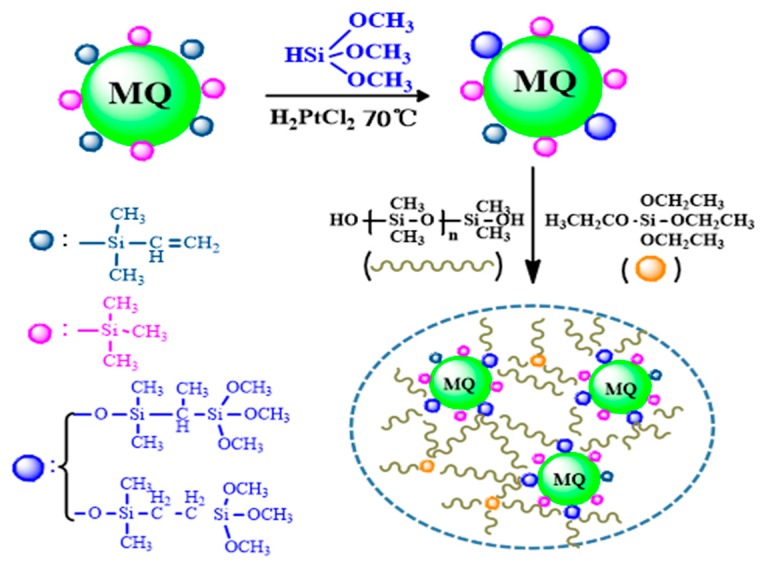
Prepared route of RTV silicone rubber using MMQ as a cross-linker.

**Figure 2 polymers-11-01142-f002:**
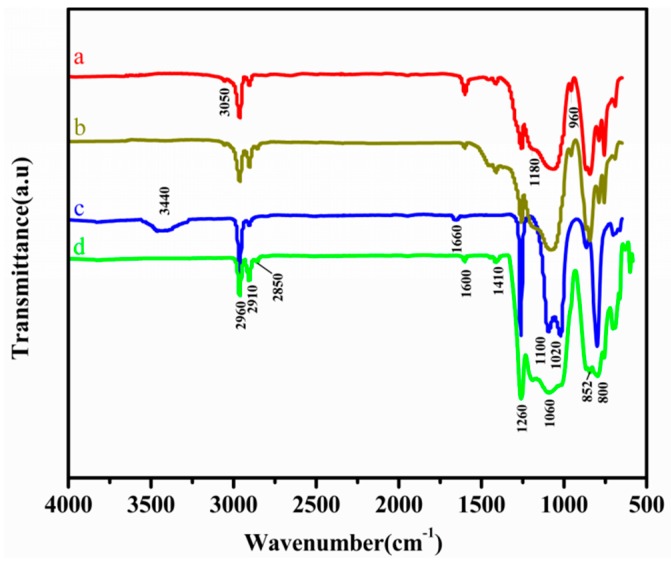
Infrared spectra of (**a**) vinyl–MQ (VMQ), (**b**) MMQ, (**c**) HPDMS and (**d**) RTV silicone rubber (MMQ-1).

**Figure 3 polymers-11-01142-f003:**
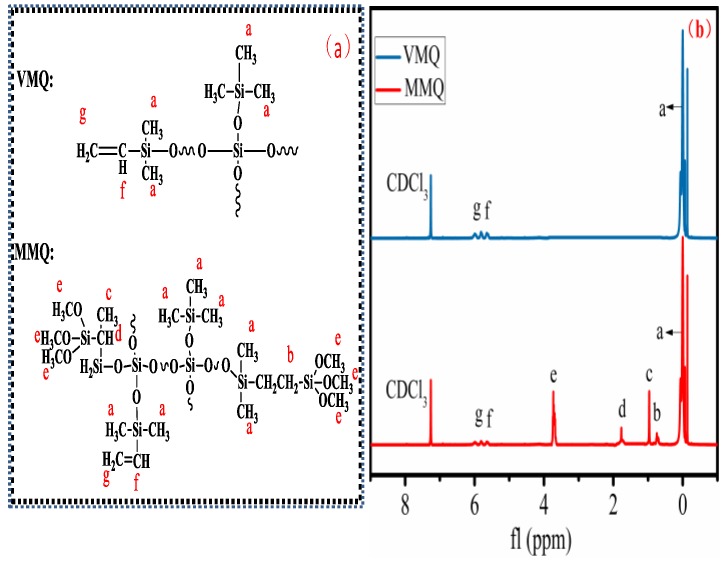
1H NMR spectroscopic analysis. (**a**) Molecular formulas of VMQ and MMQ, (**b**) 1H NMR spectra of VMQ and MMQ.

**Figure 4 polymers-11-01142-f004:**
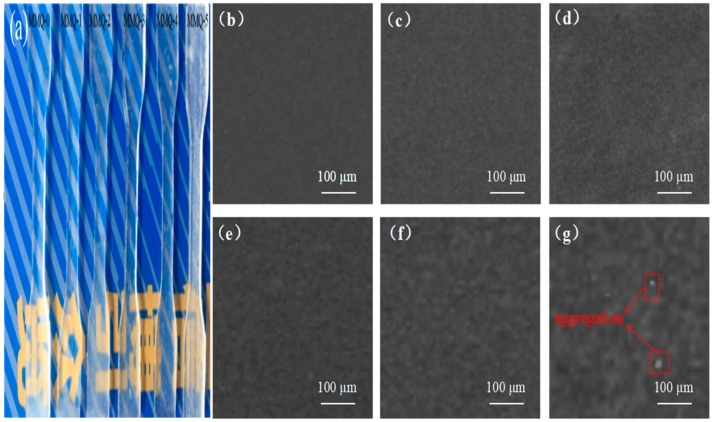
Morphology of the RTV silicone rubbers. (**a**) Photographs of samples from MMQ-0 to MMQ-5. SEM micrographs of RTV silicone rubbers. (**b**) MMQ-0, (**c**) MMQ-1, (**d**) MMQ-2, (**e**) MMQ-3, (**f**) MMQ-4 and (**g**) MMQ-5.

**Figure 5 polymers-11-01142-f005:**
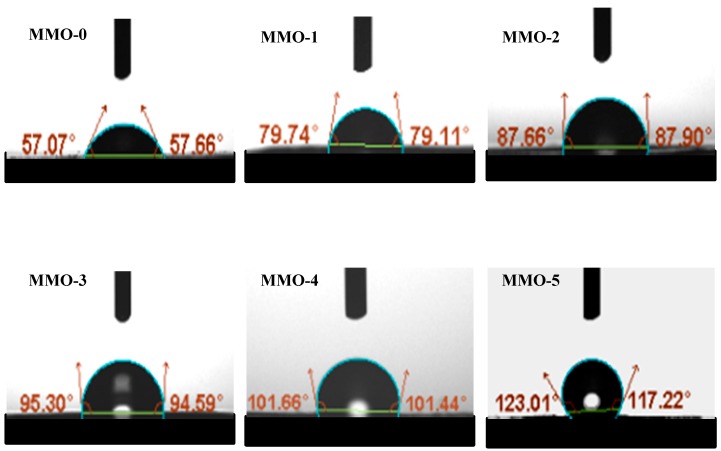
The contact angle of the RTV silicone rubber.

**Figure 6 polymers-11-01142-f006:**
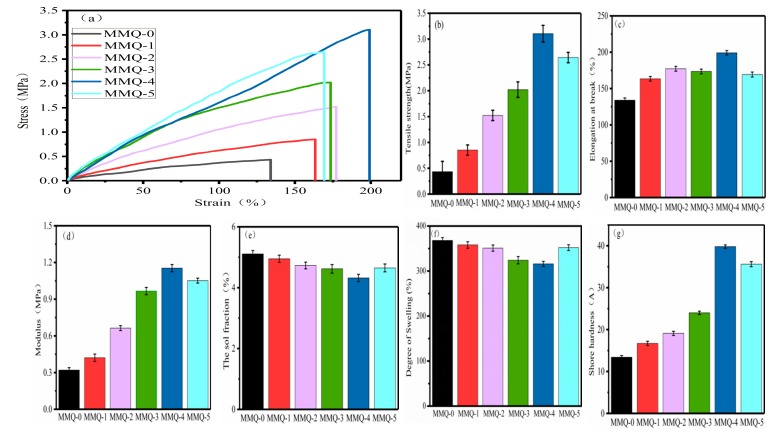
The mechanical properties of RTV silicone rubbers: (**a**) stress–strain curves; (**b**) tensile strengths; (**c**) elongation at break; (**d**) Young′s Modulus; (**e**) sol fraction; (**f**) degree of swelling; and (**g**) shore hardnesses of RTV silicone rubber.

**Figure 7 polymers-11-01142-f007:**
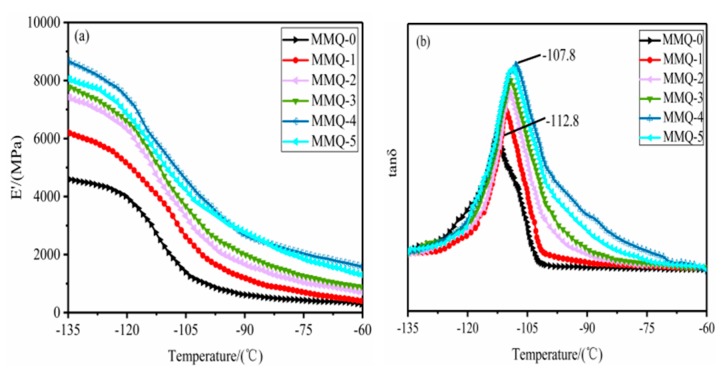
The storage modulus (**a**,**b**) and the tanδ versus temperature curves for the RTV silicone rubber.

**Figure 8 polymers-11-01142-f008:**
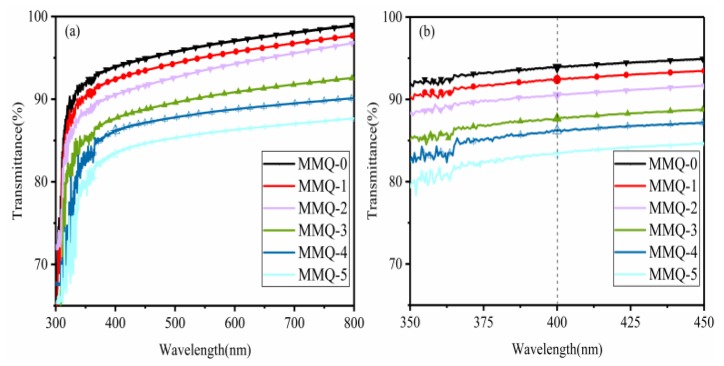
The UV-transmittance of RTV silicone rubber (**a**), and (**b**) the spectrogram of the spectra from 350 nm to 450 nm.

**Figure 9 polymers-11-01142-f009:**
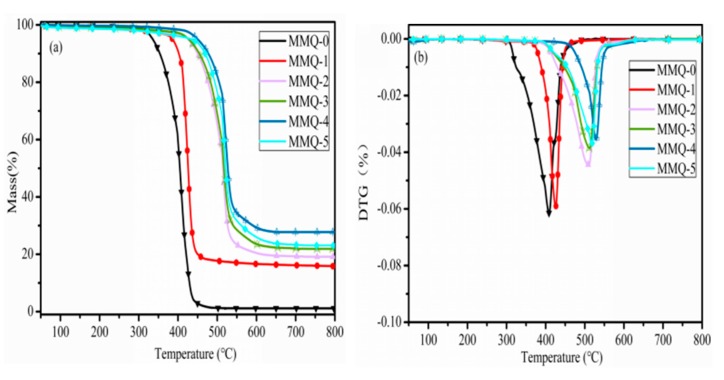
TGA (**a**) and DTG (**b**) curves of RTV silicone rubbers obtained in a nitrogen atmosphere.

**Table 1 polymers-11-01142-t001:** The amounts of raw materials of room-temperature vulcanized (RTV) silicone rubber.

Sample	HPDMS/g	MMQ/g	TEOS/g	DBTDL/μL	MMQ/wt %	Alkoxy of MMQ/mmol	Alkoxy of TEOS/mmol	Total Alkoxy /mmol
MMQ-0	50	0	3.19	200	0	0	61.37	61.37
MMQ-1	50	5.87	2.85	200	10	6.52	54.84	61.37
MMQ-2	50	13.11	2.43	200	20	14.56	46.81	61.37
MMQ-3	50	22.25	1.91	200	30	24.72	36.65	61.37
MMQ-4	50	34.15	1.22	200	40	37.94	23.43	61.37
MMQ-5	50	50.29	0.29	200	50	55.87	5.50	61.37

HPDMS: hydroxy-terminated polydimethylsiloxane; MMQ: methoxyl-capped MQ silicone resin; TEOS: tetraethyl orthosilicate; DBTDL: dibutyltin dilaurate.

**Table 2 polymers-11-01142-t002:** The important characteristic data of TG curing for RTV silicone rubbers obtained in a N_2_ atmosphere.

Sample	Temperature of 10% Weight Loss/°C	Temperature at Maximum Degradation Rate/°C	Residual Yield at 800 °C/%
MMQ-0	353.5	408.9	1.2
MMQ-1	438.5	426.3	15.9
MMQ-2	449.8	509.2	19.1
MMQ-3	454.3	510.7	21.9
MMQ-4	477.1	528.4	27.7
MMQ-5	468.8	515.4	23.1

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
