# Peer review of "Synthesis and Characterization of Room Temperature Vulcanized Silicone Rubber Using Methoxyl-Capped MQ Silicone Resin as Self-Reinforced Cross-Linker"

_polymers, 2019, doi:10.3390/polym11071142_

Round 1

Reviewer 1 Report

Dear Editor, in the present study methoxyl-capped MQ silicone resin MMQ) was synthesized by hydrosilylation of vinyl-containing MQ silicone resin and trimethoxysilane,and then, used in the condensed RTV silicone rubber as a self-reinforced cross-linker. After carefully reading I found that the paper is well organized, has a low of experiments and contains new data that could be interested for many scientists. It is a competed study and for this reason I propose to accept it for publication. Just a minor comment for the abstract, which I propose to be rewritten since the first sentence has no meaning. Also the most important findings should be added here or in the conclusions.

Author Response

Dear Reviewer,

Thank you for your comments. We have studied the valuable comments from you and tried our best to revise the manuscript which we hope meet with approval. Please download the revised version together with ourresponses to you. Welcome to offer valuable comments about our work again. Thank you.

Kind regards,

Jianye Ji

Reviewer 2 Report

The authors have not clearly indicated which catalyst was used at the condensation stage. Most likely, it was dibutyltin dilaurate, but this should be clearly indicated in the manuscript.

On page 5 of the manuscript in chapter 2.5. Mass fraction of methoxy groups in MMQ The authors describe the method for the determination of the OMe groups content in synthesized resin expressed in %. However, the results of these studies were not provided in the manuscript. The only information on the content of methoxy groups is given on page 6 of the manuscript in chapter 3.1 Structural analysis as 1.12 mmol/g. It is not clear whether this value was determined based on the procedure indicated in chapter 2.5 or determined on the basis of 1H NMR analysis results. The issue of the content of methoxy groups in the prepared MMQ resin and the method of their determination is very important from the point of view of the interpretation of the rest of the results obtained. My reservations, in this case, arise from the fact that on page 6 of the manuscript in line 224, the authors write: "So the excess molar ratio of TMOS to VMQ was used to promote reaction completely, the Choice of the molar ratio of Si-H/Si-CH=CH2 was 1.05/1 at this experiment.” However, in the 1H NMR spectra shown in Figure 3, signals characteristic for vinyl groups (5.56-6.12 ppm) are present both on the VMQ resin spectrum and subjected to the modification MMQ one, which proves their incomplete conversion. The issue of the method for the OMe groups content determination and the calculation of the degree of conversion of vinyl groups should be clarified and their numbers should be precisely determined prior to the manuscript publication.

Author Response

(The authors gave the same response as above.)

Round 2

Reviewer 1 Report

Dear editor,

In the revised manuscript the proposed comments have been taken into consideration and all necessary changes have been done in the submitted manuscript. For this reason I propose to accept it for publication.

Reviewer 2 Report

The revised manuscript is suitable for publication however, it still needs to be type edited in my opinion mainly due to the many punctuation mistakes.